# Does the Nano Character and Type of Nano Silver Coating Affect Its Influence on Calcareous Soil Enzymes Activity?

Ahmad Bazoobandi [1],*, Amir Fotovat [1],*, Akram Halajnia [1] and Allan Philippe [2]

1    Department of Soil Science, Faculty of Agriculture, Ferdowsi University of Mashhad, Mashhad 917751163, Iran
2    iES Landau, Group of Environmental and Soil Chemistry, Institute for Environmental Sciences, Koblenz-Landau University, Fortstraße 7, 76829 Landau, Germany
*    Correspondence: bazoobandi.ahmad@mail.um.ac.ir (A.B.); afotovat@um.ac.ir (A.F.)

**Abstract:** Numerous applications of silver nanoparticles (AgNPs), as well as the toxic effects of these particles on soil organisms and microorganisms, raise the question of how reasonable the entry of these nanoparticles into the environment is. Studying the behavior of these nanoparticles with soil organisms and also their effect on soil microorganisms may be the first step to finding out the answer. Structural and form differences in these nanoparticles for use in different conditions can change their behavior. The surface of these nanoparticles is covered with diverse coatings with differing surface charges affecting their fate in soil environments. Naturally, studying this aspect is essential to better understand how these particles impact the environment. In the present study, urease and dehydrogenase enzymes were used as soil health indicators to evaluate the effect of AgNPs and silver nitrate ($AgNO_3$). In order to investigate the effect of surface charge, concentration, and exposure time, three concentration levels (5, 25, 125 mg/kg soil), three different types of charged coatings (citrate (Cit), polyvinylpyrrolidone (PVP) and polyethyleneimine (PEI)) were added to the soil as a treatment and the activities of dehydrogenase (as an indicator of overall microbial activity) and urease (indicator of nitrogen cycle) were measured at three times (1 h, 1 day and 90 days) after soil contamination. The results showed that with increasing the concentration of $AgNO_3$ and AgNPs, the amount of dehydrogenase and urease activity decreased significantly. In the case of urease enzyme, nanoparticles with PEI coating (positive charge) had the greatest effect on reducing activity. In the case of dehydrogenase the opposite was true, and nanoparticles with Cit coating (negative charged) showed a greater inhibitory effect. With increasing incubation time, the amount of enzymatic activity of both types of enzymes showed less decrease, so that the greatest decrease in activity was in the first hour, then in the first day, and finally in 90 days. By comparing silver nitrate and nanoparticles it was found that the effect of AgNPs on enzymatic activity was greater than silver nitrate.

**Keywords:** Ag; $AgNO_3$; citrate; dehydrogenase; nanoparticles; polyethyleneimine; polyvinylpyrrolidone; urease

## 1. Introduction

Antibacterial [1], antifungal [2] and antiviral effects [3] of silver nanoparticles (AgNPs) have made them one of the important components of various health products. The use of nanoparticles in various products results in their entry into the environment [4], raising the question of their environmental impact, especially in soil. The entry of nanoparticles into the soil could be a possible starting point for AgNPs to enter the human food chain. AgNPs could cause a variety of diseases and cancers in humans and other plant-eating organisms [5]. To study these issues, various environmental aspects and factors affecting the behavior of these particles must be considered. In general, there are two main factors that determine the behavior of a substance including the intrinsic properties of that substance and environmental conditions [6]. In the case of nanoparticles, their size, aggregation state,

and surface properties are the most important factors in controlling their mobility and transport in environmental aqueous and terrestrial systems. These parameters determine the interactions with solutes and natural surfaces such as biofilms, algae, plants, fungi, and sediments [7]. On the other hand, in the soil environment, parameters such as ionic strength, pH, the amount of organic matter, the amount of clay, etc., are the determining factors for the behavior of soil nanoparticles [8]. Antimicrobial properties of AgNPs are one of the concerns of the scientific community about the entry of this substance into the soil, because the presence of soil microorganisms is an important factor in soil quality and health while the presence of these nanomaterials can cause the destruction of many soils' beneficial microorganisms [9], earthworms [10] and the activity of important soil enzymes [11].

Soil enzymes play a vital role in soil processes such as the food cycle and energy conversion through numerous chemical, physical, and biological reactions [12]. Given the importance of soil enzymes' activity in environmental issues and the effect of this parameter as an indicator of soil health, this parameter can be used as a tool for monitoring the effect of soil contaminants. Urease plays an important role in soil health as one of the important factors in the nitrogen cycle. Additionally, dehydrogenase is recognized as a major indicator of soil microbial activity [13]. Various researchers have suggested using soil enzymes as indicators for soil contamination, fertility, health, and maturity [14]. Among the main reasons for this approach are the close relationship of soil enzymes with organic matter and soil physical and biological properties, ease of measurement, and their rapid response to changes in soil management [15,16]. The activity of soil enzymes depends on the process and intensity of biochemical processes and is influenced by soil type, land use, vegetation, and soil management plan [17,18]. The effect of nanoparticles on soil enzyme activity is difficult to generalize because there are many factors that affect the results, including nanoparticle type, size and coating, application concentration, exposure time, and enzyme type.

In commercial products, surface coatings are usually used to prevent aggregation [19]. The coating can, moreover, increase biocompatibility or cell-specificity in biomedical applications [19]. For example, AgNPs can be coated with polyvinylpyrrolidone or citrate to enhance their stability [20] and to perform as a selective antibiotic by normalizing with ATP [21], or equipped with COOH- and $NH_2$ groups and to the surface charge from the point of view of function may be affected in imaging and drug delivery [22]. The surface functionalization of AgNPs largely regulates their bioavailability, physicochemical fate and toxicity when released into the environment [23,24]. Wu et al. [25] in nano cosmic research, noticed that polyethylene glycol AgNP has the highest overall toxicity, followed by AgNP silica, and finally amino silica-coated AgNP. The stability is different because the dissolution rate is different. These coatings are usually functional groups, surfactants, polyelectrolytes or polymers [26] and, therefore, normally have a charge and cause a positive or negative charge at the nanomaterial surface, which triggers the fate of nanoparticles [27]. Another trigger is the soil composition. Indeed, the presence of large amounts of carbonates, sulphate, bicarbonates, gypsum and other active ligands in calcareous soils can affect the transport of AgNPs as well as their aggregation and sorption. Additional to soil type and AgNP functionalization, a few studies have also recorded the noteworthy impacts of exposure time on the toxicity of AgNP in soil [28–30].

Accordingly, and considering the calcareous soil of the study area, the study of the effects of silver nanoparticle surface coatings with different charges in calcareous soils, and how they affect enzymatic activity, can have interesting results. In particular, we hypothesize that the charged cations in calcareous soils can affect the behavior of silver nanoparticles with different coatings. Because positively charged cations such as calcium and magnesium are abundant in these soils, they can accumulate negatively charged nanoparticles and reduce their effect on enzymes. Although many studies have been performed on the effects of AgNPs on soil microbial population and soil enzyme activity, their results have been contradictory. It is generally found that the effect of AgNPs on

enzymatic activity in soil depends on the type of enzyme, the time of exposure, the size of the nanoparticles, the surface coating, and the concentration. However, so far no one has studied the effects of silver nanoparticle surface coatings on the activity of urease and dehydrogenase enzymes in a calcareous soil. Although there are many studies in relation to other soils, the difference in terms of organic matter and charged cations, etc., is the reason for the present study. In this regard, in order to investigate the effect of surface coating of AgNPs and silver nitrate on the activity of urease and dehydrogenase enzymes, we designed an experiment in which the amounts of enzyme activity are measured at different times. In this experiment the soil was contaminated with AgNPs with citrate (Cit), polyvinylpyrrolidone (PVP) and polyethyleneimine (PEI) coatings as well as silver nitrate at three concentration levels (5, 25, 125 mg·kg$^{-1}$ (soil)), and the effects of these treatments on the enzymatic activity of urease and dehydrogenase in calcareous soils were measured over time.

## 2. Materials and Methods

Calcareous top-soil (0–15 cm) samples were obtained (October 2018) from a field in Khorasan Razavi Agricultural Research Education and Extension Organization (AREEO) (36°16′14″, 59°33′25″). The studied soil has been collected from plowed fields ready for wheat cultivation. Since the microbial activity is related to the physical and chemical properties of the soil, the main characteristics of the test soils were measured, and are listed in Table 1. Based on this, chemical properties such as soil pH and electrical conductivity (ECe) of the samples were measured after soil saturation for 24 h in saturated soil extract using pH meters (model METROHM 632, Herisau, Switzerland) and electrical conductivity (model JENWAY 4310) [31], equivalent calcium carbonate was measured by the [32] method, Organic carbon by [33] method, Cation exchange capacity by [34] method, and particle size distribution was measured by standard hydrometric method [35].

**Table 1.** The main characteristics of the test soil. (OC: organic carbon, CEC: cation exchange capacity).

| Parameter (Unit) | Value |
|---|---|
| pH | 7.97 |
| EC (dS·m$^{-1}$) | 0.45 |
| CEC (cmol$_c$·kg$^{-1}$) | 8.5 |
| OC (%) | 0.58 |
| CaCO$_3$ (%) | 17 |
| Sand (%) | 63 |
| Silt (%) | 13.3 |
| Clay (%) | 23.7 |
| Texture class | Sandy clay loam |

First, the study soil was air dried for 24 h, passed through a 2 mm sieve and stored in a polypropylene plastic container prior to enzymatic activity determination; soil samples were treated with 0 (control), 5, 25, 125 mg of silver nitrate/nanoparticles per kg of dry soil in powder form and then homogenized with an electric mixer (model: Set-1113). They were then incubated for three months (the moisture content was adjusted to about 60% of the water holding capacity). This was done for each of the AgNPs' treatments, and of course the enzymatic activity was measured after one hour, one day, and 90 days of incubation. The soil pH was measured for each exposure concentration and duration, but no significant differences were observed.

### 2.1. Characterization of AgNPs

AgNPs with three types of surface coatings, citrate (Cit), polyvinyl pyrrolidone (PVP) and polyethylene amine (PEI), as well as silver nitrate were purchased from the ASEPE Nano Tech (Tabriz, Iran). Transmission electron microscope (TEM) (Model EM10C-80KV, Carl Zeiss, Jena, Germany) AgNPs' images were obtained at an accelerating voltage of 150 kV. The sample was prepared by placing a droplet of the colloidal suspension

(200 mg L$^{-1}$) on a Cu mesh TEM grid and left for air drying. The particle size distribution of AgNPs was performed using dynamic light scattering (DLS) method using Zetasizer Nano ZS system (Malvern Instruments Ltd., Malvern, UK).

### 2.2. Urease Activity

To measure the activity of urease enzyme, 5 g of soil was first treated with 0.2 mL of toluene to stop microbial activity. Then, 9 mL of Tris buffer (Tris hydroxymethy-laminomethane, pH = 7) and 1 mL of 0.2 M urea solution were added to it and incubated for 2 h at 37 °C. After incubation, 35 mL of KCl-Ag$_2$SO$_4$ solution (2.5 m/L relative to KCl and 100 mg/L relative to Ag$_2$SO$_4$) were added, cooled to room temperature, and then soil suspension diluted in 50 mL with KCl-Ag$_2$SO$_4$ solution; then the amount of ammonium released into the soil mixture was determined by steam distillation [36]. To do this, 20 mL of the suspension was transferred to a distillation vessel and 0.2 g of magnesium oxide was immediately added to it and placed in a Kjeldahl device. Then the distillated ammonia was titrated with sulfuric acid 0.005 N. The control sample did not receive urea at the beginning of incubation, and after 2 h of incubation and adding KCl-Ag$_2$SO$_4$ solution 1 mL of urea solution was added and the amount of ammonium was measured immediately. Using the difference between the amount of control ammonium nitrogen and the main treatment, the urease activity was reported in terms of milligrams of ammonium released per gram of dry soil in two hours of incubation (mg(NH$_4^+$-N) g$^{-1}$ (soil) 2 h$^{-1}$) [37].

$$\frac{mL \times 0.07 \times 50 \times 1000 \times 100}{20 \times 5 \times \%dm} = \text{mgN·g}^{-1}\text{dm·2h}^{-1}$$

where mL = The volume of sulfuric acid; 0.07 = Conversion factor; 50 = Extract volume; 1000 = Conversion factor; 20 = The volume of the extract to be measured; 5 = Initial weight of soil; $\frac{100}{dm\%}$ = Dry soil mass conversion factor.

### 2.3. Dehydrogenase Activity

The modified [38] method was used to measure the activity of soil dehydrogenase. First, 5 mL of 0.6% 2,3,5-triphenyltetrazolium chloride solution (TTC) as a substrate (optimal substrate concentration in tris buffer between 0.1 %to 2%; the concentration is determined according to the type of soil) was added to 5 g of soil and was incubated at 25 °C for 16 h. After incubation, 25 mL of acetone was added to the soil sample to extract the triphenylformazan (TPF) and stirred with a shaker (Edmund buhler model: ks-15, Germany) for 2 h in dark conditions. After stirring for 2 h, the mixture was passed through Whatman 42 filter paper and the amount of light absorption of the samples at 546 nm was measured by spectrophotometer (Dynamica model:Halo xb-10, Iran). The amount of enzyme activity was reported as micrograms of triphenylformazan per gram of dry soil at 16 h of incubation (µg TPF g$^{-1}$ soil 16 h$^{-1}$).

$$\frac{(S - C) \times 100}{5\,dm\%} = \text{µgTPF·g}^{-1}\text{dm·16 h}^{-1}$$

where S = Average TPF in the sample; C = Average TPF in the control; 5 = Initial weight of soil; $\frac{100}{dm\%}$ = Dry soil mass calculation factor.

### 2.4. Random Forest

To compare the effects of variables with each other using a multivariate statistical model, we used random forest model here. For this (R package ranger), the time of incubation, type of treatments (coating) and concentration of treatments were taken as environmental variables (predictor variables) and the activity of urease and dehydrogenase as output variables (response variables). Initially, 70% of the data (36 sample with 3 replicate) were used randomly for model training and 30% for testing. Our random forest was based on the question of which parameters have the greatest impact on the activity of enzymes:

Treatment (coating), Time, or concentration? Then we could not only conclude that the factors have an effect, but also find out which factors are the most important for predicting the effect. Fifty different random initial splits were produced to evaluate the stability of the RF model. The premutation method was used to determine the important parameter.

### 2.5. Statistical Analyses

The statistical design was completely randomized with factorial arrangement and data were considered as 3 replicates; data were analyzed using JMP software (Pro, Version 16, SAS Institute, Cary, North Carolina, USA), and Tukey test and random forest (RF) were carried out using R Studio (Version 5501.9.0.0, Posit, Vienna, Austria).

## 3. Results

### 3.1. Soil Samples

As can be seen in Table 1, some physical and chemical characteristics of the soil samples are presented. According to the results the studied soil has high amounts of lime and low amounts of organic matter.

### 3.2. Characterization of AgNPs

The characteristics of the AgNPs used are shown in Figure 1. According to the figure, it is clear that the average size of particles with different coatings is almost equal. TEM images show a good dispersion of silver nanoparticles, so that nanoparticles with a smaller size have a homogeneous distribution and a spherical shape, and a few larger particles are slightly more irregular and show a very small amount of entanglement.

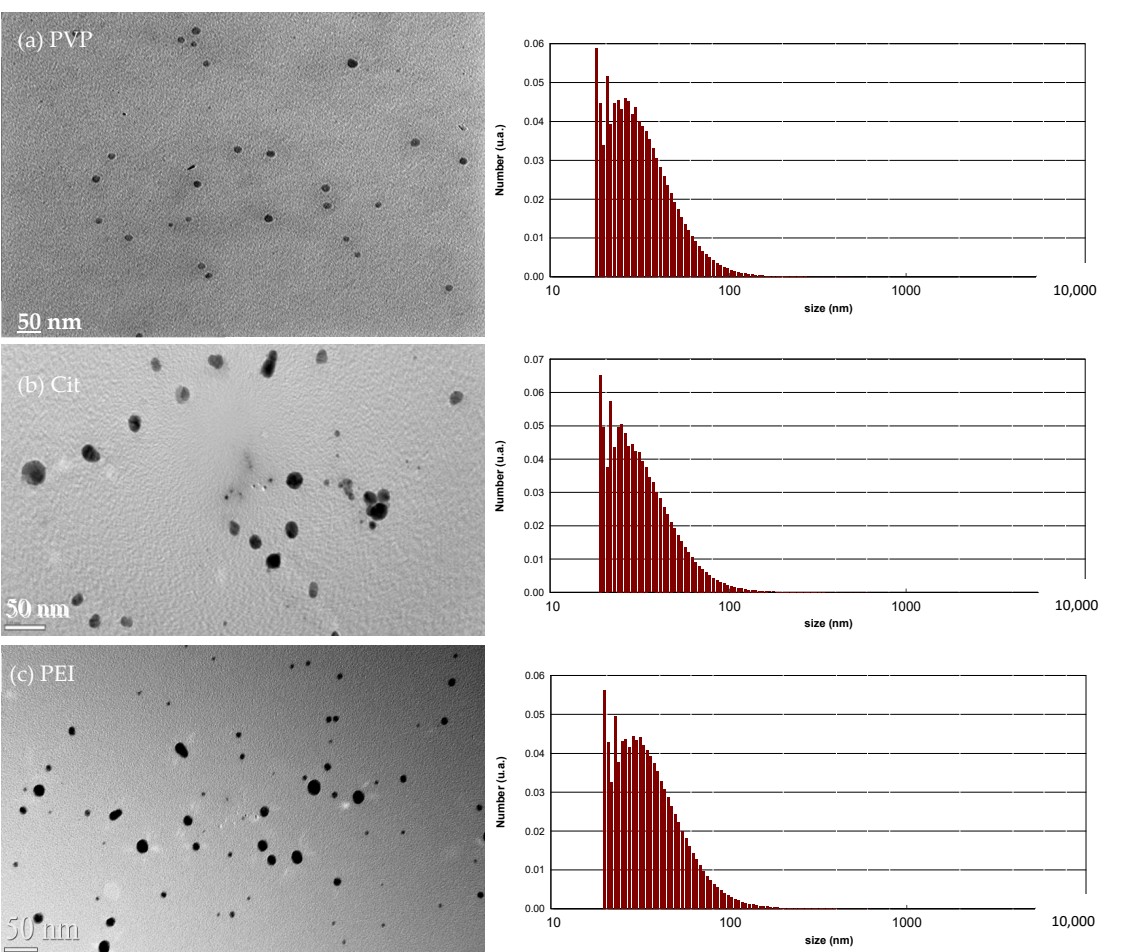

**Figure 1.** Size distribution and Transmission electron microscope (TEM) image of different AgNPs: (**a**) AgNP-PVP; (**b**) AgNP-Cit; and (**c**) AgNP-PEI.

### 3.3. Urease Activity

As shown in Figure 2, the results are determined by concentration, time and type of silver particles. According to the results and as expected, silver in the form of nitrate and nanoparticles with different coatings have reduced the activity of urease enzyme. The activity of urease has been reported based on the percentage of activity compared to the control. With increasing concentration in different treatments, the amount of activity has decreased. As the results show, with increasing time from one hour to 90 days the amount of enzymatic activity has increased sharply. A closer look at the results shows that the greatest reduction in urease activity of the highest dosage was in the treatment of AgNPs with PEI coating about 90%. Moreover, by comparing the silver nitrate treatment and AgNPs it is found that the level of urease activity in contact with AgNPs has decreased much more. However, among AgNPs silver with PEI and Citrate coating had a more severe reduction effect (90% and 85% at time 1 h). In addition, the PVP decreased less than the others (about 70%). As shown in the figure, at the time of one hour in all treatments only the effect of silver nitrate at concentrations of 5 and 25 was significantly reduced. However, at 90 days all treatments show a significant decreasing trend. At a concentration of 125, only silver nitrate at the times of one day and 90 days shows a significant decreasing effect.

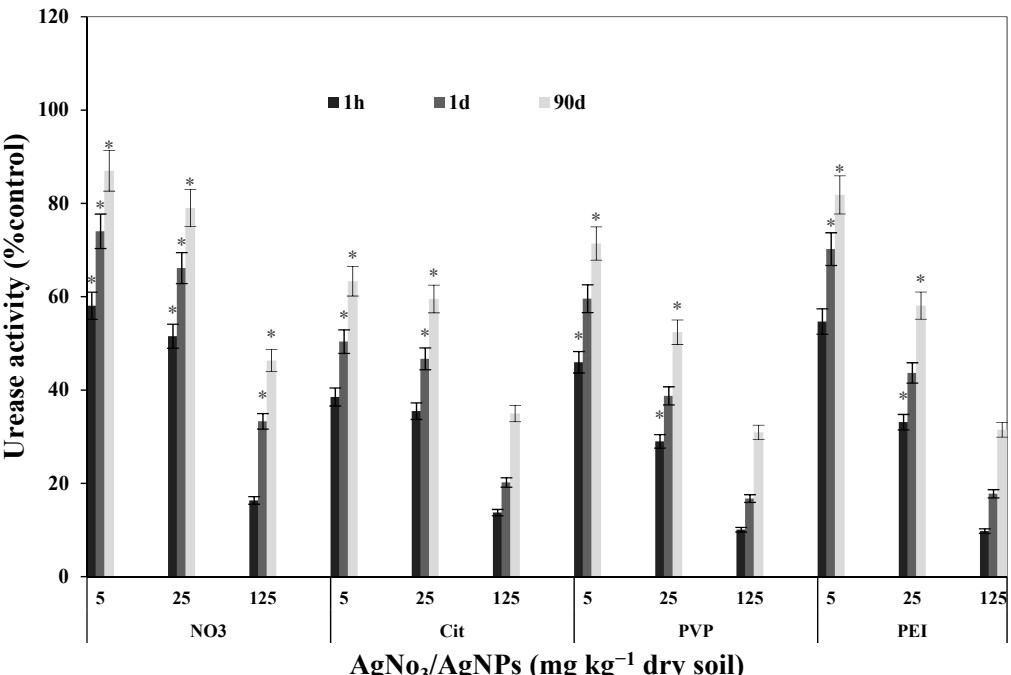

**Figure 2.** Activities of urease as a percentage of the mean of the control (no Ag/AgNPs added) treatments exposed to 5, 25, 125 mg AgNO$_3$/AgNPs (Cit, PVP, PEI coated) kg$^{-1}$ dry soil after 1 h, 1 d and 90 d. Values are the means of three replicates $\pm$ standard deviation. The asterisk (*) denotes significant differences with the control ($p < 0.05$).

### 3.4. Dehydrogenase Activity

The effects of silver nitrate and AgNPs on dehydrogenase activity are shown in Figure 3. As is obvious from the figure, this effect of treatments on dehydrogenase activity was not decreasing. For example, about effects of concentration, in some treatments a slight incremental effect can be seen. According to the results, it is clear that the sensitivity of dehydrogenase enzyme to AgNPs is higher than silver nitrate, which in low concentrations has a higher activity of this enzyme. The highest inhibitory effect of dehydrogenase enzyme was observed in AgNPs with citrate coating treatment, even at lower concentration. An interesting issue in the available results is the recovering of dehydrogenase activity over time for all treatments. In particular, the inhibitory effect of PVP-AgNPs completely disappeared after 90 days, whereas the recovery was incomplete for Cit-AgNPs and PEI-

AgNPs even after 90 days. In 90-day treatments, the greatest reduction effect is related to PEI-coated nanoparticles, which is seen at a concentration of 125. As expected, with increasing the concentration of experimental treatments the level of inhibition of enzyme activity has also increased. In some treatments like AgNPs-Cit 125, AgNPs-PVP 125 and AgNPs-PEI 125 the sensitivity of the enzyme at one day is remarkably greater than at one hour and 90 days.

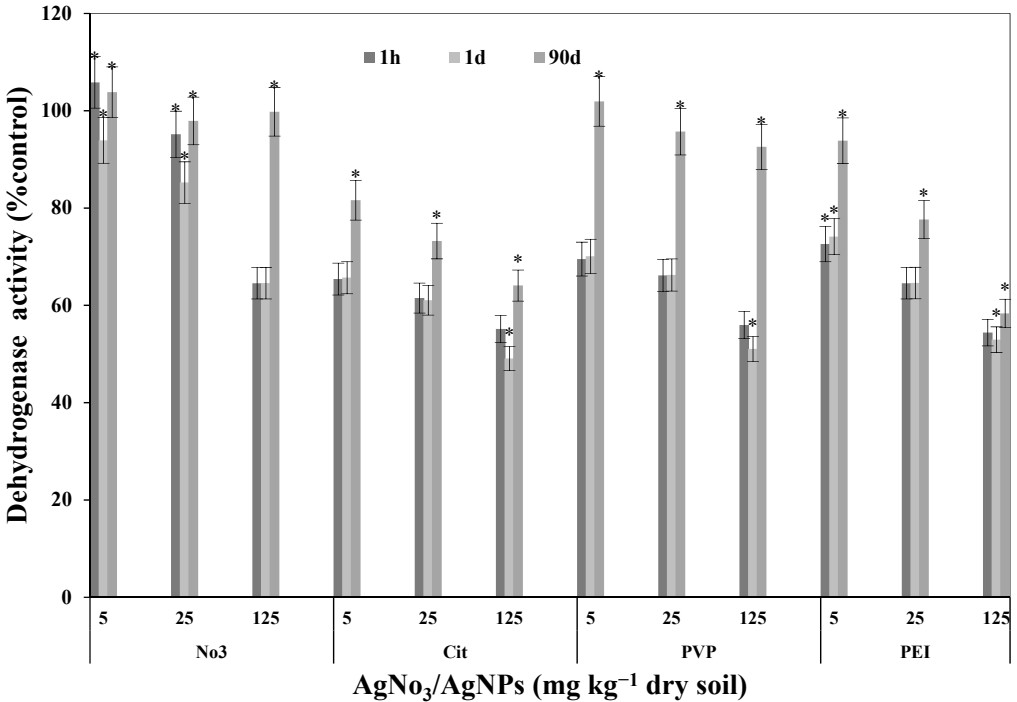

**Figure 3.** Activities of Dehydrogenase normalized by the mean of the control (no Ag/AgNPs added) replicates for samples treated with 5, 25, and125 mg $AgNO_3$/AgNPs (Cit, PVP, PEI coated) $kg^{-1}$ dry soil after 1 h, 1 d, and 90 d. Values are the means of three replicates ± standard deviation. The asterisk (*) means significantly less than the control ($p < 0.05$).

*3.5. Random Forest*

As the results show, the random forest models generally perform well in prediction, although they perform much better for urease than for dehydrogenase. After running 50 models, the aggregated results in boxplots (Figure 4) show the importance for each input parameter, or how important the given parameter is for making a proper prediction in the model. For urease, the concentration is clearly the leading parameter followed by the time, whereas the coating does not seem to play a significant role here. In other words, we could predict the activity of the particles against the enzyme by knowing the concentration and the time of exposure only. According to the density diagram, it can be concluded that the model has predicted the parameter affecting the urease activity with high accuracy. However, the results for dehydrogenase are different, as shown in Figure 4 (density diagram) where the accuracy of the model in predicting the most effective parameter was less than urease. Moreover, the effect of different parameters on dehydrogenase activity is not so different from each other. For dehydrogenase, time is the most important parameter and the concentration and coating have the same effects.

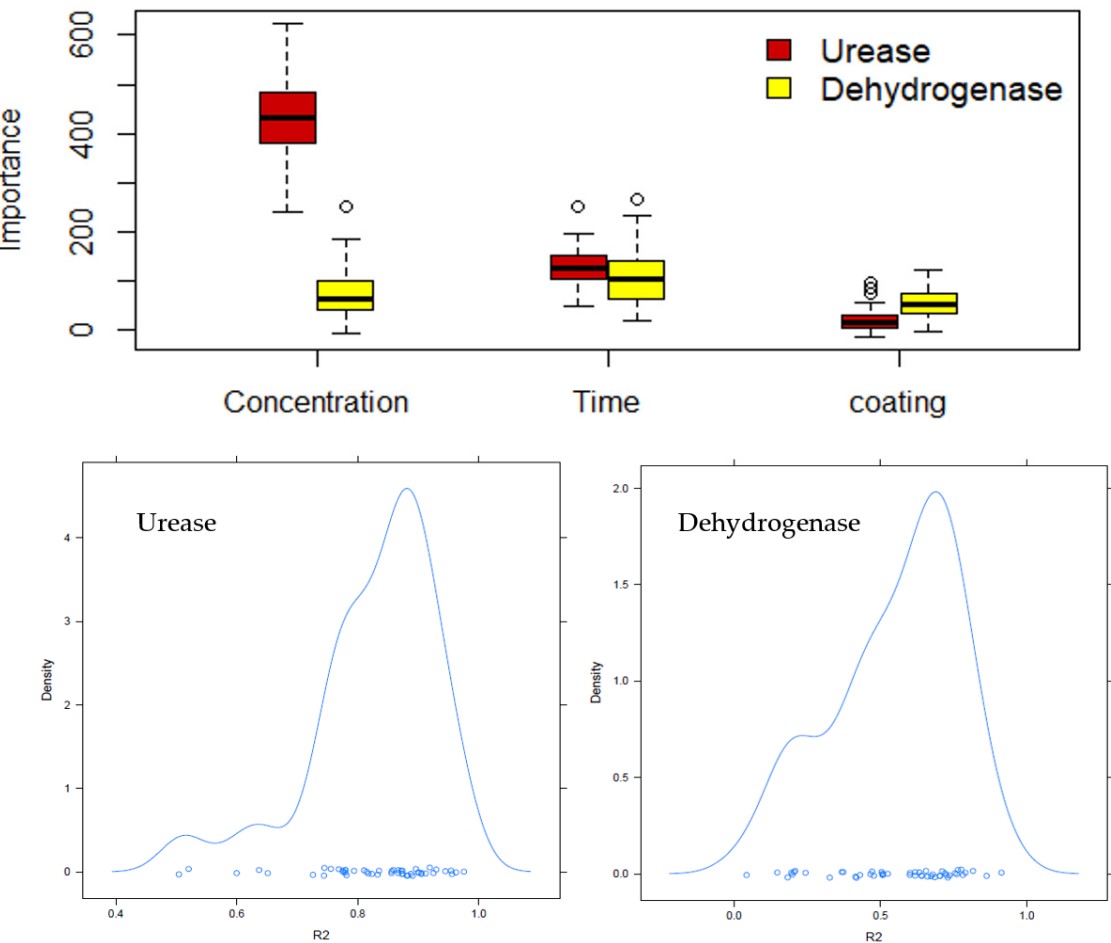

**Figure 4.** Random forest model and Variable importance for the urease and dehydrogenase activity and density plot of the mean R-squared value of 50 model.

## 4. Discussion

As stated in previous studies, we expected AgNPs and silver nitrate to have an inhibitory effect on the activity of soil enzymes, but how this works in calcareous soil was the issue that led us to design this experiment. As in previous studies, our results showed that the effect of AgNP on the activity of soil enzymes was highly dependent on the type of enzyme, time of exposure, coating agents and concentration of AgNPs [18,39,40]. In this study, as an important indicator for soil health, nitrogen cycle and heavy metal pollution, the activity of urease and dehydrogenase in exposure to AgNPs and $AgNO_3$ was discussed. Urease is mostly extracellular, cytosolic enzyme catalyzing the hydrolysis of urea to ammonia and carbon dioxide [41]. Dehydrogenase is an important indicator to reflect microbial activities, also an imperative enzyme included within the transformation of soil organic matter, and can endorse dehydrogenation of carbohydrates and organic compounds, playing the critical part of transitional carrier of hydrogen [13].

After exposure to the $AgNO_3$ and AgNPs, urease activity was inhibited following a significant dose–response relationship. AgNPs' urease activity inhibition at high concentrations was higher than for $AgNO_3$. Ref. [40] observed that urease activity was sensitive to AgNP and $AgNO_3$, and a clear dose–response relationship was found between urease activity and AgNPs' levels. After applying $AgNO_3$ and AgNPs to the soil, the urease activity decreased to about 90% at 125 mg·g$^{-1}$ and did not recover after 90 days. This suggests that AgNPs significantly inhibited the cycle of nitrogen in the soil ecosystem. Ag$^+$ can inhibit urease activity by blocking sulfhydryl groups (SH groups) [42,43], thus acting as a non-competitive inhibitor of urease by changing the enzyme's structure. For AgNPs the same mechanism could apply after particle dissolution. Additional mechanisms specific to

AgNPs were reported by [44], such as the reduction in the enzymatic effect of urease after its absorption on the nanoparticles' surface, which blocked the accessibility of the urease active site. Another possibility is that the adsorbed protein's conformation changes at the level of the secondary structure or at the active site of individual enzymes, hence reducing its activity.

Interestingly, the highest inhibitory effect for urease is related to positively charged AgNPs at higher concentrations. These nanoparticles could have a higher interaction with the enzyme and reduce enzymatic activity. On the other hand, due to the high concentration of divalent cations in calcareous soils it can be pointed out that some cations are adsorbed on the surface of these nanoparticles with negative and neutral charges [45] inducing aggregation, and hence reducing the surface available for the enzyme. Furthermore, aggregation reduces the mobility of nanoparticles in the soil, which could indirectly decrease its biological activity.

Over time the inhibitory effect increased for all treatments, indicating a significant recovery of the biological system. The recovery rate varied with treatment and concentration. The recovery could be explained by the aging and consequent deactivation of the AgNPs over time. Previous studies have shown that the exposure time of soil enzymes to AgNPs can alter the magnitude of the observed effect [39]. Aging provides more opportunities for soil components such as cations, clay, and soil organic matter (SOM) in the area to come into contact with and interact with nanoparticles. This can improve the ability of nanoparticles to immobilize through different modes of absorption, thus reducing the bioavailability of the particles exposed to enzymes. Saturation of the NP's surface could be another explanation as additional enzymes would not be able to adsorb. Another transformation of interest could be sulfidation as there is sulfur in the proteins and in humic substances. S–Ag bonds are thermodynamically very stable and would reduce the NP's activity on the long term. Sulfidation is quite a slow process, and this would explain why it takes a long time for the activity to recover. Sorption and aggregation, however, are quite fast processes and it would not take 90 days to be completed. The current findings on urease and dehydrogenase agree with previously reported studies showing that prolonging the contact time of the nanoparticles with the soil reduces the effect of the nanoparticles. Furthermore, the recovery may be related to the capacity of soil microorganisms to metabolize and endure nanoparticles, as well as their capacity to adjust to stressors through recovery and quality translocation.

In the present study, the inhibitory effect of AgNPs on dehydrogenase was less than urease, in accordance with [40], but in contrast to [46] who pointed out that dehydrogenase activity is more sensitive to inhibition than urease. These contrasting findings might be explained by the extent of biotic and abiotic variables affecting Ag contaminants. For example, the physicochemical properties of the soils or the differing composition of the soil microbial communities could affect the results. Ref. [47] used soils belonged to the same pedological series that were different in properties due to the land use patterns such as farmland, meadow, woodland, wasteland, housing and roadside over an area of $667 \, \text{km}^2$, and reported that soil properties had significant effects on the activities of soilborne enzymes in heavy metal polluted soils. At the highest level of inhibition, AgNPs reduced dehydrogenase activity by about 50%. Citrate-coated nanoparticles show the highest impact with a strong effect even at low concentrations. As [13] reported, the effect increases with the nanoparticle concentration. It is interesting to note that unlike urease enzyme, AgNPs with negatively charged coatings have the greatest effect on dehydrogenase activity. Dehydrogenase is an intracellular enzyme, and this shows that in the presence of negatively charged nanoparticles the number of living organisms has decreased more compared to other AgNPs and AgNO$_3$. The most interesting point about that is that the NPs have to be integrated into the cells before they can interact with the enzyme, unless we just observe a general biocide effect which indirectly reflects in a decreasing in enzymatic activity overall. However, the fact that the urease is more inhibited than hydrogenase could suggest that the NPs are interacting directly with the urease, whereas for hydrogenase this is not possible.

The recovery from the dehydrogenase activity's inhibition was barely significant after one day, but high after 90 days, and even almost complete after 90 days for PVP; it could be that the effect of being not so strong is that it is easier for the microorganisms to recover compared to urease.

In many studies researchers have attributed the effect of AgNPs to soil enzymes due to the release of silver ions from nanoparticles, believing that the effect of silver nitrate as a source of silver ions in the environment can be more severe than AgNPs. This is not the case in the present study in which the inhibitory effects of AgNPs on urease and dehydrogenase clearly depend on the coating of the nanoparticles and, therefore, cannot be merely due to the release of silver ions. Indeed, the inhibitory effect of AgNPs on enzyme activity was higher than for $AgNO_3$, regardless of concentration and incubation time [40], moreover, detailed that AgNPs have a more prominent effect on soil enzyme activities than Ag ions because of changes within the enzyme's conformation exposing the active site when bound to the surface of the nanoparticle [48]. It is also proposed by some researchers that the adsorption of $Ag^+$ soil colloids or the precipitation/complexation of $Ag^+$ with humic acid, phosphate, chloride, or sulfides, for instance, reduces their effect compared to AgNPs [11,49]. In this regard, some researchers have attributed the decrease in enzymatic activity due to treatment with nanoparticles to their heavy metal characteristics. They believe that these properties can block the binding sites of the enzyme surface [50]. In a study conducted by [11] on the effect of AgNPs and $Ag^+$ on the activity of urease and alkaline phosphatase enzymes, it was found that AgNPs have a more limiting effect than silver nitrate.

Previous studies have suggested that although the mechanism of action of AgNPs and silver nitrate is still unclear [51,52], several factors, such as the intrinsic characteristics of nanoparticles [52], the type of coating [53], the concentration [18], the size [54], and of the surface charge of these nanoparticles, as well as the characteristics of the enzyme and the soil composition, can affect the behavior of these particles. The results of the present study, according to the random forest model, show that the type of coating of AgNPs in the case of dehydrogenase enzyme can affect the enzymatic activity as much as the concentration and the incubation time, although in the case of urease enzyme this effect is less than the concentration and incubation time. In addition, this issue should be considered in future studies because it can directly and indirectly affect the behavior of nanoparticles in the environment [55,56].

### 5. Conclusions

This study was performed to investigate the effect of AgNPs with different coatings, as well as silver nitrate, at three concentration levels on the enzymatic activity of dehydrogenase and urease at different incubation times. Both enzymes showed a reducing effect when exposed to experimental treatments. This inhibitory effect of experimental treatments on the activity of enzymes increased with increasing concentration and decreased over time. The sensitivity of urease to experimental treatments was significantly higher than dehydrogenase. The effect of AgNPs was greater than that of silver nitrate on the enzymatic activity of both enzymes. In general, according to the random forest model, it was observed that the result of concentration has the greatest effect among the studied treatments, but it is interesting to note that in some cases the effect of the type of coating can be as important as the concentration. Although the toxic effects of AgNPs do not seem to be very dangerous to humans at concentrations found in the environment, the indirect effects of these pollutants can affect the food cycle and ultimately human life; therefore, understanding the behavior of these nanoparticles in the environment is essential.

**Author Contributions:** Conceptualization, writing—original draft preparation, A.B., Conceptualization, writing—original draft preparation and supervision A.F.; advisor, A.H.; software, writing—review and editing, A.P. All authors have read and agreed to the published version of the manuscript.

**Funding:** This research is part of Ph.D. thesis of the Ferdowsi university of Mashhad and received not external funding.

**Institutional Review Board Statement:** Not applicable.

**Informed Consent Statement:** Not applicable.

**Data Availability Statement:** The authors declare that the data supporting reported results of this study are available within the article.

**Conflicts of Interest:** The authors declare no conflict of interest.

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
