# Peer review of "Does the Nano Character and Type of Nano Silver Coating Affect Its Influence on Calcareous Soil Enzymes Activity?"

_coatings, doi:10.3390/coatings12121968_

Round 1
Reviewer 1 Report
Manuscript of ‘Does the Nano character and type of Nano silver coating affect 2 its influence on calcareous soil enzymes activity?’ is well written. However, some corrections are required to improve the quality of the manuscript. In addition to that, grammatical errors are detected in the manuscript. Kindly send for proofreading service.
Major correction is required for the formatting of the manuscript. For example, the references and the abstract.
For further improvement:
Abstract:
1. Please follow the format of the journal accordingly before submitting the manuscript.
2. Please omit the word “is” at the end of the sentence for the background.
3. One hour, one day and 90 days- please synchronize the numbering. Better put 1 hour, 1 day and 90 days.
4. Decreased sharply- not scientific. Author is advised to change to significantly. Maybe the authors can include some data to see how significant the result in term of number or percentage.
Introduction: The introduction is too long. Kindly rewrite and make it concise.
Methodology:
1. Table 1 should be in the result section.
2. Figure 1 should be in the result section. Also, please add the scale bar for the image.
3. Please provide the formula of urease and dehydrogenase activities.
Discussion:
Irregularity of font is detected.

Author Response
Manuscript of ‘Does the Nano character and type of Nano silver coating affect 2 its influence on calcareous soil enzymes activity?’ is well written. However, some corrections are required to improve the quality of the manuscript. In addition to that, grammatical errors are detected in the manuscript. Kindly send for proofreading service.
Major correction is required for the formatting of the manuscript. For example, the references and the abstract.
For further improvement:
Abstract:
- Please follow the format of the journal accordingly before submitting the manuscript.
Done!
- Please omit the word “is” at the end of the sentence for the background.
It changed.
- One hour, one day and 90 days- please synchronize the numbering. Better put 1 hour, 1 day and 90 days.
Done
- Decreased sharply- not scientific. Author is advised to change to significantly. Maybe the authors can include some data to see how significant the result in term of number or percentage.
Done
Introduction: The introduction is too long. Kindly rewrite and make it concise.
We tried to remove some sentences. But the first paragraph of the introduction section is related to the statement of the problem and the importance of the present study. The next paragraph is about the importance of soil enzymes. The next paragraph is about the importance of the surface tension of silver nanoparticles. And in the last paragraph, why and the necessity of doing the study. With all due respect, changing or removing any of these parts can cause problems in the structure and understanding of the necessity of the study
Methodology:
- Table 1 should be in the result section.
Done!
- Figure 1 should be in the result section. Also, please add the scale bar for the image.
Done!
- Please provide the formula of urease and dehydrogenase activities.
ml= The volume of sulfuric acid
0.07= Conversion factor
50=Extract volume
1000=Conversion factor
20=The volume of the extract to be measured
5=Initial weight of soil
=Dry soil mass conversion factor
And
S= Average TPF in the sample
C= Average TPF in the control
5= Initial weight of soil
= Dry soil mass calculation factor
Were added
Discussion:
Irregularity of font is detected.
The font irregularity fixed
Reviewer 2 Report
The topic discussed by the manuscript of Ahmad et.al is could be an interesting topic to the general community, however, there are major questions that needs to be addressed before the manuscript could be considered for publication in Coatings. As such, the manuscript needs some corrections, based on the following suggestions:
1. Adherence to the journal's referencing style is very important, and this must be corrected in the article.
2. Interesting how the abstract was structured, although it is not the conversional way, it however fail to incite the interest for the reader to continue reading, as the summary from the title question isn't answered. Therefore, it must be improved.
3. Authors should avoid the use of acronyms, such as in line 95. It should be 'For example or for instance' in stead of 'e.g'.
4. From the summary of the introduction, it is apparent that the study is on the microbial impact of the coated AgNPs, focusing mainly on the urease and dehydrogenase enzymes in calcareous soils. As such the introduction must be focused on the this topic, in stead on the general usage of coated Ag nanoparticles.
5. Upon selection of the top-soil, was the usage of natural or commercial fertilizers taken into consideration?
6. Figure 1 quality must be improved, especially the particle size distribution curves. Also, what was the magnification on the TEM micrographs?
7. What were the size distributions of the Ag nanoparticles under different coatings?
8. Font of lines 304-310 must be corrected.
9. Based on the results, the authors compared the activity of Ag+ in AgNO3 to the nanoparticles coated with polymers. Is this the correct way to approach the study, since they are basically comparing the activity of the ionic Ag to that of metallic Ag?
10. High resolution of the AgNPs are needed since not only the small size of the metal nanoparticle, but also the shape of the nanoparticle affect its activity.
11. Taking into account the different sizes of the nanoparticles, authors must explain how each affect the particle size.
12. Could authors explain in detail the various interactions of the coating to the inhibition activity of the different enzymes? The interaction the ionic Ag with the enzymes must also be discussed in detail, especially in the calcareous soil media.
13. Figure 4 must be labelled the caption must correspond to the figure as well.
14. Generally, what was the outcome of the study: in what form is silver toxic to the microbial enzymes? What governed the effect of the coating such that the inhibition is observed?
Author Response
The topic discussed by the manuscript of Ahmad et.al is could be an interesting topic to the general community, however, there are major questions that needs to be addressed before the manuscript could be considered for publication in Coatings. As such, the manuscript needs some corrections, based on the following suggestions:
- Adherence to the journal's referencing style is very important, and this must be corrected in the article.
The references style changed to the coating style
- Interesting how the abstract was structured, although it is not the conventional way, it, however, fails to incite the interest of the reader to continue reading, as the summary from the title question isn't answered. Therefore, it must be improved.
It changed
- Authors should avoid the use of acronyms, such as in line 95. It should be 'For example or for instance instead of 'e.g'.
It changed
- From the summary of the introduction, it is apparent that the study is on the microbial impact of the coated AgNPs, focusing mainly on the urease and dehydrogenase enzymes in calcareous soils. As such the introduction must be focused on this topic, instead of the general usage of coated Ag nanoparticles.
The first paragraph of the introduction section is related to the statement of the problem and the importance of the present study. The next paragraph is about the importance of soil enzymes. The next paragraph is about the importance of the surface tension of silver nanoparticles. And in the last paragraph, why and the necessity of doing the study. With all due respect, changing or removing any of these parts can cause problems in the structure and understanding of the necessity of the study.
- Upon selection of the topsoil, was the usage of natural or commercial fertilizers taken into consideration?
Using or not using fertilizers is not important in this matter because for all samples it was the same, but we didn’t use any fertilizer at all.
- Figure 1 quality must be improved, especially the particle size distribution curves. Also, what was the magnification on the TEM micrographs?
The figure changed
- What were the size distributions of the Ag nanoparticles under different coatings?
To avoid the size effects the size of nanoparticles was the same 25-35 nanometers for all nanoparticles as stated at fig1.
- Font of lines 304-310 must be corrected.
It changed
- Based on the results, the authors compared the activity of Ag+ in AgNO3 to the nanoparticles coated with polymers. Is this the correct way to approach the study, since they are basically comparing the activity of the ionic Ag to that of metallic Ag?
The idea of comparing the NO3 with the nanoparticles was comparing the nanoparticles with the bulk which is usual in other studies.
- High resolution of the AgNPs is needed since not only the small size of the metal nanoparticle but also the shape of the nanoparticle affects its activity.
According to TEM figures, the shape of nanoparticles is mostly spherical and the effect of the shape of nanoparticles can be ignored.
- Taking into account the different sizes of the nanoparticles, authors must explain how each affects the particle size.
We tried using the same size of nanoparticles to remove the size effects
- Could authors explain in detail the various interactions of the coating to the inhibition activity of the different enzymes? The interaction of the ionic Ag with the enzymes must also be discussed in detail, especially in the calcareous soil media.
Due to the different nature of enzymes, their effectiveness from external stress can be different, for example, in the issue of intracellular and extracellular enzymes, it can be pointed out that an external factor affects an enzyme, if it needs to enter the cell, it usually has to travel a longer path, but in extracellular enzymes, this path is shorter and the effect happens faster. Regarding the silver ion in calcareous soil, it should also be mentioned that the retention time of silver in ionic form is so long that it can affect the activity of the enzyme or not. This is an issue that we are currently investigating in a new study.
- Figure 4 must be labeled the caption must correspond to the figure as well.
The label of Fig 4. Changed to: Random forest model and Variable importance for the urease and dehydrogenase activity(a) and density plot of the mean R-squared value of 50 models..
- Generally, what was the outcome of the study: in what form is silver toxic to the microbial enzymes? What governed the effect of the coating such that the inhibition is observed?
According to the results, the coatings weren’t the important parameter in both enzymes and it was reported that the concentration is playing the main role.

Reviewer 3 Report
The manuscript “Does the Nano character and type of Nano silver coating affect its influence on calcareous soil enzymes activity?” reports an interesting study on behavior of silver nanoparticles with soil organisms and their effect on soil microorganisms.
1. Line 223-224, the figure 2 reveals that PVP coated has similar reduction as Cit and PEI coated. Authors should illustrate results in percentage change.
2. Dehydrogenase activity should also be exemplified with percentage change in data.
3. Line 20, AgNO3 should be corrected as AgNO3
4. Line 95, E.g, should be deleted.
5. Table 1. EC extension electric conductivity should be mentioned in the table legend.
Author Response
Response to Reviewer3 Comments
Point 1
Line 223-224, figure 2 reveals that PVP coated has a similar reduction as Cit and PEI coated. Authors should illustrate results in percentage change.
It changed at lines 184-185
Point 2
Dehydrogenase activity should also be exemplified with the percentage change in data.
The requested changes added at lines 198, 201
Point 3
Line 20, AgNO3 should be corrected as AgNO3
DONE.
Point 4
Line 95, E.g, should be deleted.
Done
Point 5
Table 1. EC extension electric conductivity should be mentioned in the table legend.
Done.

Reviewer 4 Report
In this manuscript the authors present an experimental work to study the effects of silver nanoparticle surface coatings on the activity of urease and dehydrogenese in soil. The effect of surface coating of AgNPs and AgNO3 on the enzymtic activity of urease and dehydrogenase at different time (1h, 1day and 90 days) was studied. The authors concluded that the sensitivity of urease was higher that for dehydrogenase, and the concentration has the greatest effect. They also showed that in some cases, the effect of the type of coating was as significant as the concentration.
This manuscript is well written and structured. It has some merits and it could be interesting for readers interested in agriculture and environmental sciences. In the opinion of the reviewer, in this study the coating aspects were marginal, even, this does not affect its significance and the quality of this investigation. Anyhow, there are some deficiencies in the manuscript that the authors should address to improve the quality of the investigation.
11) The title of the manuscript is given as a question. Rare are the papers for which the title is a question. The authors are recommended to used more informative title.
22) The abstract should be written as a paragraph, as it is the standard in the coatings Journal.
33) As the manuscript is submitted to coatings, the manuscript did not clearly describe how the coatings were obtained? what kind of coating process was used? Probably, this is the main weakness of this manuscript.
44) The authors used statistical analysis and random forest method. In the manuscript these aspects are not clearly stated. No sufficient information was given on the number of samples used to be statistically analyzed.
Author Response
Comments and Suggestions for Authors
In this manuscript the authors present an experimental work to study the effects of silver nanoparticle surface coatings on the activity of urease and dehydrogenese in soil. The effect of surface coating of AgNPs and AgNO3 on the enzymtic activity of urease and dehydrogenase at different time (1h, 1day and 90 days) was studied. The authors concluded that the sensitivity of urease was higher that for dehydrogenase, and the concentration has the greatest effect. They also showed that in some cases, the effect of the type of coating was as significant as the concentration.
This manuscript is well written and structured. It has some merits and it could be interesting for readers interested in agriculture and environmental sciences. In the opinion of the reviewer, in this study the coating aspects were marginal, even, this does not affect its significance and the quality of this investigation. Anyhow, there are some deficiencies in the manuscript that the authors should address to improve the quality of the investigation.
11) The title of the manuscript is given as a question. Rare are the papers for which the title is a question. The authors are recommended to use a more informative title.
It changed to Effect of the Nano character and type of Nanosilver coating on calcareous soil enzymes activity
22) The abstract should be written as a paragraph, as it is the standard in the coatings Journal.
Done!
33) As the manuscript is submitted to coatings, the manuscript did not clearly describe how the coatings were obtained. what kind of coating process was used? Probably, this is the main weakness of this manuscript.
As we mentioned in line 129, the silver Nanoparticles with different coatings were purchased from ASEPE Nano Tech and we didn’t synthesize them.
44) The authors used statistical analysis and the random forest method. In the manuscript, these aspects are not clearly stated. No sufficient information was given on the number of samples used to be statistically analyzed.
In lines 161-175 we clearly mentioned how we used the random forest and did the statistical analysis, but it is worth mentioning that we used all samples (3 coating agents and NO3 silver nanoparticles, 3 concentration levels, and 3 times of enzyme activity measurement) that we had from the enzyme activity to do the statistical analysis and random forest (36 samples with 3 replicate). The change was also added in line 165.

Round 2
Reviewer 1 Report
Manuscript of ‘Does the Nano character and type of Nano silver coating affect 2 its influence on calcareous soil enzymes activity?’ is well written and all the corrections have been incorporated. However, some minor corrections are required to improve the quality of the manuscript.
1. Table 1: Please add the word "(unit)" after the word parameter.
2. Figure 2 and Figure 3: Please omit the space between the figures and the figure legends.
3. Consider revising the figure legends for Figure 1,2,3,and 4. Add more explanation in the figure legends as it should be self explanatory.
Author Response
- Done
- the spaces removed
- we couldn't understand what should we add to the legend! we think the captions are the explanation for the figures and legends should be simple.
Reviewer 2 Report
The corrections have been addressed with great satisfaction.
Author Response
Thank you for your efforts.
there isn't any comment to replay.